# Urban–Rural Distinction or Economic Segmentation: A Study on Fear and Inferiority in Poor Children’s Peer Relationships

**DOI:** 10.3390/healthcare10102057

**Published:** 2022-10-17

**Authors:** Shencheng Wang, Baochen Liu, Yongzheng Yang, Liangwei Yang, Min Zhen

**Affiliations:** 1Department of Social and Ecological Studies, Party School of the Central Committee of CPC (National Academy of Governance), Beijing 100089, China; 2School of Law, Shandong Jianzhu University, Jinan 250101, China; 3School of Public Administration and Policy, Renmin University of China, Beijing 100872, China; 4School of Public Policy and Management, Tsinghua University, Beijing 100190, China; 5School of Foreign Studies, South China Agricultural University, Guangzhou 510642, China

**Keywords:** children, fear of peers and inferiority, urban–rural distinction, economic segmentation, China

## Abstract

Peer relationships play an important role in the growth of children. This study offers insights about feelings of fear and inferiority in children’s peer relationships. Based on a national survey, the 2018 Construction for Social Policy Support System for Urban and Rural Poor Families in China, initiated by the Ministry of Civil Affairs, and using multiple regression models and a structural equation model, this study discusses whether and how having a rural household registration or being from a poor (*dibao*) family has an isolation effect on fear and inferiority in children’s peer relationships. The research findings indicate that children with a rural household registration or those from a *dibao* family are at a disadvantage in peer interactions. Moreover, rural resident identity has an indirect effect on children’s fear of peers and inferiority, mainly through psychological resilience, anxiety and depression, and mobile phone dependence. Being from a *dibao* family directly influences children’s fear and inferiority in their peer relationships; it also indirectly influences fear of peers and inferiority through psychological resilience. This study suggests that more attention should be paid to fear of peers and inferiority in rural children or children from a *dibao* family.

## 1. Background

Previous studies have shown that peer relationships have a unique and irreplaceable role in the social and emotional development of children. These relationships impact the healthy development and social adaptation of children’s social ability, cognition, emotion, self-conception, and personality [1,2]. Peer relationships and family environment are the two core systems of children’s personality formation and socialization [3].

Peer relationships can be positive or negative. Early research primarily examined peer relationships from two perspectives: peer acceptance and friendship. Peer acceptance is a one-way structure of common opinions, reflecting the attitudes group members have toward individuals, such as likes or dislikes, acceptance, or exclusion. Friendship is an individual-oriented two-way structure, reflecting the emotional connection of individuals. With the development of research, scholars have gradually paid attention to difficulties in children’s peer relationships [4], such as peer rejection [5] and peer victimization [6]. However, aside from peer exclusion and peer victimization, fear of peers and sense of inferiority are the individual subjective feelings of fear and inferiority in peer interactions, which are associated with social self-perception [7]. From a field perspective, peer exclusion and peer aggression are mainly concentrated in the middle and end of peer communication processes, while fear of peers and inferiority are mainly concentrated in the front end. Fear of peers and inferiority may be the major factors behind children’s resistance to social participation and social integration. They might be detrimental to the production of prosocial behavior and could lead to aggressive behavior against others [8]. It is necessary to conduct an in-depth study on the antecedents of fear of peers and inferiority, which will increase the understanding of why some children fall into an adverse situation in peer relationships; such a study will also identify the effective measures needed to help children build and maintain good peer relationships.

Therefore, this study focused on fear of peers and inferiority in Chinese children. In reality, the reasons for children’s fear of peers and inferiority may be multifaceted. However, in China, it is important to pay particular attention to two unique social backgrounds that are very important for children’s growth. First, there is an urban–rural separation effect in children’s development in China [9]. The differences in the geographic regions and the human environments between urban and rural China lead to a gap between urban and rural children’s environmental adaptation and interpersonal communication skills [10]. Second, the rapid development of China’s economy has produced serious economic divisions between people. Some vulnerable groups have fallen into poverty and have also suffered from social exclusion in terms of political participation and interpersonal communication [10]. Children from poor families, with low social skills and a lack of communication experience, are prone to psychological problems, such as feelings of inferiority [11]. In the study discussed in this paper, we aimed to determine whether a rural household registration or being from a poor family (*dibao* family) has an isolation effect on fear and inferiority in children’s peer relationships and to investigate the specific mechanisms underlying this.

## 2. Research Questions and Analytic Frameworks

Although fear of peers and inferiority in children’s peer interactions are influenced by socio-economic status (SES), some empirical studies have suggested that the effect of SES on children’s peer relationships is very weak [12]. These controversies suggest that the effects of SES on children’s fear of peers and inferiority may be influenced by mediating mechanisms. Therefore, the present study aimed to examine the association between SES and fear of peers and inferiority among children in China. Moreover, it explored the intermediary mechanisms of the SES stratification constituted by urban–rural distinction and economic segregation that affects fear of peers and inferiority from the perspective of social stratification.

### 2.1. Socio-Economic Stratification and Peer Fear and Inferiority

In China, urban–rural distinction and economic segmentation constitute two important aspects of socio-economic stratification. Urban–rural distinction has become an undeniable fact in China. Due to the long-term existence of the country’s household registration system, there is a clear distinction between urban and rural areas. Moreover, this system has a comprehensive effect on urban–rural distinction [13], and a disadvantage in peer interactions has been confirmed, thus becoming a typical problem of children’s development [9]. Studies have indicated that, due to urban–rural geographical and cultural differences, a certain degree of difference also exists among college students, in terms of environmental adaptation and interpersonal communication [14]. In comparison to rural students, urban students have better interpersonal communication skills [15]. Regarding group cooperation, rural children get along with others significantly better than children from villages and towns [16]. Therefore, this study proposes:

**Hypothesis** **1.**
*Urban–rural distinction has a negative effect on children’s fear of peers and inferiority.*


In addition to urban–rural distinction, acute economic segmentation arises from China’s rapid economic development. Long-term emphasis on economic efficiency and neglect of social justice have led to a significant gap between rich and poor, as well as a wide social class divide. Economically, some vulnerable groups of society not only fall into poverty, but also suffer all-around social exclusion with regard to political participation and interpersonal communication [13]. Among the associated disadvantages, social exclusion resulting from interpersonal communication is particularly harmful to the growth of poor children and students. Family poverty not only causes and aggravates students’ psychological burden, it also negatively impacts their communication needs leading to a low level of social skills and a relative lack of contact experience. They have a high probability of confronting many psychological problems, such as an inferiority complex, impacting their interpersonal communication skills [11]. Students with family economic difficulties are a high-risk group for psychological poverty [17], and there is a gap between poor and non-poor undergraduates [18]. Therefore, this study proposes:

**Hypothesis** **2.**
*Economic segmentation has a negative effect on children’s fear of peers and inferiority.*


### 2.2. Psychological Resilience

Resilience research began in the 1970s. Some scholars believe that resilience refers to an individual’s ability to cope with changes and stressful events in a healthy way [19], while others emphasize that resilience is a process of reintegration. When children encounter serious sources of pressure, they can return to normal with the support of protective factors [20]. Studies have indicated that resilience helps diminish children’s depressive symptoms and enables them to initiate peer relationships and cultivate more of them. Children with resilience are more popular among their peers. Consequently, they enjoy better interpersonal relations and social support networks [21] and have relatively more stable and effective social support resources [22]. 

However, other studies have suggested that SES indirectly reflects the abundance of resources that individuals can mobilize and utilize [23,24]. Individuals with lower SES may incur more health costs in maintaining psychological resilience and they may exhibit poorer mental health [25]. For example, children from rural areas typically have a lower SES, which in turn reduces their level of psychological resilience [26]; this may increase their fear and feelings of inferiority in their peer interactions. Therefore, this study proposes:

**Hypothesis** **3.**
*Psychological resilience partially mediates the relationship between urban–rural distinction and children’s fear of peers and inferiority.*


**Hypothesis** **4.**
*Psychological resilience partially mediates the relationship between economic segmentation and children’s fear of peers and inferiority.*


### 2.3. Anxiety and Depression

Anxiety and depression are the commonly diagnosed psychological disorders among children. Anxiety is a group of mental disorders characterized by anxiety and fear, often accompanied by severe depression or other personality disorders. There is a statistically significant correlation between anxiety and depression [27]. Studies have shown that adolescent anxiety and depression has a significant negative correlation with peer relationships. The higher the degree of anxiety and depression is, the worse the child’s peer relationships [28]. Children with a higher level of anxiety and depression have poorer social functioning, less classmate support, and less social acceptance in social communication [29]. High anxiety and social insecurity will increase the risk of children’s low-quality friendships and peer abuse, and a low level of social support and peer relationships will further deepen children’s psychological distress, such as anxiety and depression [30]. 

Further research has shown that anxiety and depression are closely related to SES. Individuals with lower SES showed stronger anxious depression than individuals with higher economic and social status [31]. Therefore, this study proposes:

**Hypothesis** **5.**
*Anxious depression partially mediates the relationship between urban–rural distinction and children’s fear of peers and inferiority.*


**Hypothesis** **6.**
*Anxious depression partially mediates the relationship between economic segmentation and children’s fear of peers and inferiority.*


### 2.4. Mobile Phone Dependence

Attention overload theory considers that individual psychological resources are limited, and the maintenance of target information depends on the number of available psychological resources. The failure of sustained attention comes from limited psychological resources [32]. When individuals with high dependence on mobile phones input a large number of cognitive resources into those devices, they reduce the resources that should have been allocated to other personal activities. Consequently, excessive dependence on mobile phones will lead to children’s cognitive failure in social communication as well as many adverse psychological characteristics, such as stress susceptibility and low self-evaluation [33]. Studies have found that, in a group of young people with mobile phone addiction, the negative factors impacting peer relationship quality are more significant than the positive factors. Furthermore, the higher the degree of mobile phone addiction, the more negative the impact is on the quality of peer relationships [34]. Social phobia is significantly associated with the risk of smartphone addiction in young people. Individuals with psychosocial problems, such as social phobia and loneliness, prefer mobile devices rather than face-to-face communication because online communication can reduce anxiety [35]. 

Research on mobile phone dependence has shown that it is closely related to economic and social status. Students from *dibao* families have higher levels of cell phone addiction than students from non-*dibao* families [36,37,38]. Therefore, this study proposes:

**Hypothesis** **7.**
*Mobile phone dependence partially mediates the relationship between urban–rural distinction and children’s fear of peers and inferiority.*


**Hypothesis** **8.**
*Mobile phone dependence partially mediates the relationship between economic segmentation and children’s fear of peers and inferiority.*


## 3. Methods

### 3.1. Participants

The data used in this study were collected by the Peking University Chinese Social Sciences Survey Center in 2018, extracted from a survey project called Chinese Social Policy Support System for Vulnerable Families (CSPSS). The Ministry of Civil Affairs of the People’s Republic of China appointed and funded the Institute of Social Science Survey (ISSS) at Peking University to deliver the related project and organize a research team to write the report. It is a national large-scale sample survey project supported by the Chinese Ministry of Civil Affairs, aiming to be representative of the vulnerable Chinese families targeted by the government’s social assistance program. Using stratified sampling methods, the project adopted the computer-assisted personal interviewing (CAPI) method to investigate more than 1800 villages in 29 provinces from July 2018 to September 2018. The project has compiled three questionnaire databases: disability, the elderly, and children. Among them, parents and their children were interviewed for the children questionnaire, which included detailed information of the demographic, socio-economic, health, learning, and psychological and social interactions of the respondent parents and their children (aged 8–16 years). The respondent parents and children had to complete separate questionnaires without communicating their opinions with each other. The children were required to answer the questionnaires about children’s psychological health and school performance. If a child needed help during the procedure, an interviewer read and explained the questions. If the parents of the children (such as left-behind children in a rural area) were not at home when the interviewers were visiting, the questionnaires for parents could be also completed through a telephone survey. The database has 3342 samples, including 991 samples of urban poor families (*dibao* families), with 1032 urban non-*dibao* families; and 543 samples of rural *dibao* families, with 776 rural non-*dibao* samples. After deleting the missing and abnormal values in the database, 3334 observations were finally obtained.

### 3.2. Measurements

#### 3.2.1. Dependent Variable

To evaluate the children’s fear of peers and inferiority, we used the 10-item Fear of peers and Inferiority Scale (PFIS). The participants answered items (e.g., You feel afraid if you do something you have never done before in front of other students) on a 4-point rating scale ranging from 1 = completely disagree to 4 = completely agree. A mean score was computed to yield the composite score, and higher scores indicated higher fear of peers and inferiority. A cumulative score was created by adding the responses of all 10 indicators (ranging from 10 to 40). The higher the total score of the fear of peers and inferiority subscale, the higher the level of fear and inferiority in peer interactions, and the more negative the self-perception. In our study, the Cronbach’s alpha coefficient for the PFIS was 0.8357, demonstrating good internal consistency.

#### 3.2.2. Independent Variables

To evaluate the existence of economic segmentation and urban–rural distinction, we used two dummy variables: *dibao* family (0 = no; 1 = yes) and urban family (0 = rural family; 1 = urban family). It should be noted that, in China, families receiving *dibao* are often at the bottom of the economic status hierarchy, which can be considered to be the poorest group. 

#### 3.2.3. Mediating Variables

In this study, psychological resilience, anxiety and depression, and mobile phone addiction are the mediator variables. 

We used the Child and Youth Resilience Measure Scale (CYRM-R) to measure children’s psychological resilience. This scale was developed by Professor Michael Ungar et al. by integrating the results of 35 researchers from 11 countries and 14 regions on psychological resilience in 2009 [39]. The scale consists of 28 items, including three dimensions: individual level, relative level, and social and cultural level. They are evaluated on a five-point Likert scale, with a total score ranging from 28 to 140 points. A higher score indicates a better level of psychological resilience. In our study, the Cronbach’s alpha coefficient for the CYRM-R (psychological resilience) was 0.9045, indicating good internal consistency.

We used the Revised Child Anxiety and Depression Scale (RCADS 25) to measure the respondents’ depression tendencies. RCADS 25, which includes two dimensions (depression and anxiety), is a revised children’s anxiety and depression scale tailored for children and adolescents ranging in age from 8 to 18. The RCADS 25 uses a four-point Likert scale, with 1 representing “never” and 4 representing “always” [40]. A cumulative score (ranging from 24 to 91) was obtained by adding the responses of all 25 indicators. A higher score indicates a higher degree of anxiety and depression. In our study, the Cronbach’s alpha coefficient for the RCADS 25 was 0.8583, demonstrating good internal consistency.

To assess the tendency of mobile phone addiction, we used the Chinese version of the self-report 17-item Mobile Phone Addiction Index (MPAI), which was based on the English version of the MPAI. MPAI consists of four dimensions of mobile phone addiction: inability to control cravings, feeling anxious and lost, withdrawal/escape, and productivity loss. The participants answer items (e.g., You feel anxious if you have not checked for messages or switched on your mobile phone for some time) on a 5-point rating scale ranging from “1 = not at all” to “5 = always” [41]. A cumulative score (ranging from 14 to 83) was obtained by adding the responses of all 17 indicators. A higher score indicated a stronger tendency toward mobile phone addiction. In our study, the Cronbach’s alpha coefficient for the MPAI was 0.8528, demonstrating good internal consistency.

#### 3.2.4. Covariates

Based on the existing studies on the factors influencing peer relationships [42], the present study included three sets of covariates: school characteristics, family characteristics, and personal characteristics. School characteristics mainly included three variables: key school (*zhongdianxuexiao*), public school, and boarding school (*jisuxuexiao*); three of them are dummy variables (0 = no; 1 = yes). Family characteristics mainly included six variables: whether the parents are alive (1 = both; 0 = either or neither), whether the parents are divorced (0 = no; 1 = yes), if the parents quarrel (0 = never or rarely; 1 = occasionally or often), family gatherings (0 = never; 1 = several times a year; 2 = once a month; 3 = two or three times a month; 4 = several times a week: 5 = every day), parent-child communication (0 = never or occasionally; 1 = always or often), and parents’ beating and scolding (0 = never or occasionally; 1 = always or often). Personal characteristics mainly included five variables: gender (0 = female; 1 = male), only child (0 = no; 1 = yes), health status (0 = bad; 1 = moderate or good), physical disability (0 = no; 1 = yes), and student leader (0 = no; 1 = yes).

### 3.3. Analytical Strategies

Stata 14.0 was used as the data analysis tool for this study. First, we used the t-test to check the differences in the characteristics of two groups of participants (urban vs. rural and *dibao* vs. non-*dibao*). Then, a multiple regression model was used to examine the impact of economic segmentation and urban–rural distinction on the respondents’ fear of peers and inferiority. Finally, we used the structural equation model method of maximum likelihood with default values for model estimation. 

## 4. Results

### 4.1. Descriptive Analysis

Table 1 shows the descriptive analysis results of the core dependent variables. The average score of fear of peers and inferiority of all children was 19.942. The average score was higher for rural children (20.746) than urban children (19.415). The average score was higher for children from *dibao* families (20.381) than children from non-*dibao* families (19.570).

### 4.2. Analysis of the Multiple Regression Model

To enhance the robustness of the statistical results of the independent variables, the independent variables and three sets of control variables were gradually put into a series of multiple regression models, as shown in Table 2. Model 1 reflects the regression results when only the independent variables are included. Model 2 reflects the regression results when the independent variables and school level control variables are included. Model 3 shows the regression results when the independent variables and the school and personal level control variables are included. Model 4 shows the regression results when the independent variables and the school, personal, and family level control variables are included. With the gradual inclusion of the school, personal, and family characteristic variables, the R^2^ of the model gradually increased, indicating that the fitting degree of the model was increasingly higher. Moreover, in Model 4, the variance inflation factor (VIF) results were all lower than in Model 2 (specific results are not listed), indicating that there was no multicollinearity issue among the explanatory variables.

In Model 1, both the independent variables—of whether the child is from a *dibao* family and whether the child holds an urban household registration—passed the significance test at the 1% level. The data show that the score of fear of peers and inferiority of urban children was 1.399 points lower than that of rural children. The score of fear of peers and inferiority of children from *dibao* families was 0.916 points higher than that of children from non-*dibao* families. The results demonstrate that peer interactions were impacted by urban–rural distinction and from barriers arising from basic living allowances. In comparison to urban children and those with better economic conditions from non-*dibao* families, rural children and children from *dibao* families faced more communication barriers and had stronger fear of peers and inferiority in peer interactions.

In Model 2, both independent variables again passed the significance test at the 1% level. The children’s fear of peers and inferiority was still closely related to household registration and family economic conditions. Moreover, among the school characteristic variables, a public school or not and a boarding school or not were significantly associated with fear of peers and inferiority. The data show that the score of fear of peers and inferiority of children in public schools was 0.729 points higher than that of children in private schools. The score of fear of peers and inferiority of children in boarding schools was 1.074 points higher than that of children in the control group. The control variable of a key school or not was found to have no significant correlation with children’s fear of peers and inferiority.

Like Model 1 and Model 2, in Model 3, both independent variables passed the significance test at the 1% level. The results demonstrate that children’s fear of peers and inferiority in peer interactions were still closely related to household registration and family economic conditions. Consistent with Model 2, the control variables of a public school or not and a boarding school or not passed the significance test. Moreover, the family characteristic variables of parents’ quarrels, family gatherings, parent-child communication, and parents’ beating and scolding all passed the significance test.

In Model 4, both independent variables were significantly associated with fear of peers and inferiority, showing the same result as the other three models. Consistent with Model 2 and Model 3, the control variables of a public school or not, a boarding school or not, gender, parents’ quarrels, family gatherings, parent-child communication, and parents’ beating and scolding all showed the same significance. Personal characteristics, such as gender, only child or not, and a student leader or not, were significantly associated with fear of peers and inferiority.

Based on the descriptive analysis and multiple regression results, Hypothesis 1 and Hypothesis 2 were supported. Specifically, the score of fear of peers and inferiority was significantly higher for rural children than for urban children. The score of fear of peers and inferiority was significantly higher for children from *dibao* families than for children from non-*dibao* families. The results demonstrate that urban–rural distinction and economic segmentation impacted the children’s peer interactions.

### 4.3. Results of the Structural Equation Model

The multiple regression models demonstrated that urban–rural distinction and acute economic segmentation had an impact on the children’s peer interactions. Children with a rural household registration or those from *dibao* families suffered more from fear of peers and inferiority. However, the models cannot explain how the two factors led to a higher level of children’s fear and inferiority in peer interactions. To identify the mechanisms, this study used the structural equation model method of maximum likelihood with default values for model estimation based on the literature review and the research hypotheses. In comparison to the multiple regression analyses based on OLS (ordinary least squares), the structural equation model enabled us to conduct a path analysis more efficiently. Table 3 shows the model estimation results based on unstandardized regression coefficients.

The three mediator variables all had a significant direct effect on the score of fear of peers and inferiority of children’s peer interaction. Specifically, the score of fear of peers and inferiority decreased by 0.070 points for each point increase in children’s psychological resilience. The score of fear of peers and inferiority increased by 0.284 points for each point increase in children’s anxiety and depression. The score of fear of peers and inferiority increased by 0.048 points for each point increase in children’s mobile phone dependence.

This study found that the independent variable of whether the child is from a *dibao* family had a significant direct effect on the score of fear of peers and inferiority of children’s peer interaction; it also affected fear of peers and inferiority through the mediating mechanism of psychological resilience. The psychological resilience score was 2.180 points lower for children from a *dibao* family than for children from non-*dibao* families. However, whether the child was from a *dibao* family did not have a significant effect on the degree of anxiety and depression or on mobile phone dependence. Therefore, the influencing mechanism of a *dibao* family on children’s fear and inferiority in peer interactions was the synthesis of the direct effect brought about by basic living allowances and the indirect effect brought about by the mediating variable of psychological resilience. Thus, Hypothesis 4 was supported, but Hypothesis 6 and Hypothesis 8 were not.

This study found that the independent variable of whether the child holds an urban household registration did not have a significant direct effect on children’s fear of peers and inferiority in peer interactions. However, this variable had significant effects on children’s psychological resilience, anxiety and depression, and mobile phone dependence (all passed the significant positive test). Therefore, this variable had an indirect effect on children’s fear of peers and inferiority through the three mediating variables. Specifically, the psychological resilience score was 2.040 points higher for urban children than for rural children. Moreover the anxiety and depression score was 1.609 points lower for urban children than for rural children and the mobile phone dependence score was 1.061 lower for urban children than for rural children. These three mediating mechanisms jointly strengthened the urban–rural distinction in children’s peer interactions. Thus, Hypothesis 3, Hypothesis 5, and Hypothesis 7 were supported.

Figure 1 shows the model path diagram based on standardized regression coefficients, which allows us to understand the influencing mechanisms of urban and rural areas and basic living allowances on children’s fear of peers and inferiority more intuitively.

## 5. Conclusions and Policy Suggestions

### 5.1. Conclusions

Using data from the CSPSS national survey, this study found that urban–rural distinction and economic segmentation have an impact on children’s fear of peers and inferiority in China. First, rural children suffer much more from fear of peers and inferiority than do urban children. Hypothesis 1 was supported. In China, compared to urban children, rural children have a higher probability of becoming left behind children and they are more likely to have the characteristics of imbalance, sensitive personality, psychological isolation, inferiority and discord. Second, children from *dibao* families are more vulnerable to fear of peers and inferiority than those from families without basic living allowances. Hypothesis 2 was also supported. Because of welfare stigma, children from *dibao* families are at a disadvantage in peer interactions.

This study also examined the mechanisms of the relationship between children’s fear of peers and inferiority and urban–rural distinction and economic segmentation. The findings show that urban or rural household registration has no direct effect on children’s fear of peers and inferiority, but a rural resident identity indirectly makes rural children suffer more from fear of peers and inferiority by affecting their psychological resilience, anxiety and depression, and mobile phone dependence. Hypotheses 3, 5, and 7 were supported. Similar to previous research [23,24,25,26,31,36,37,38], this study also found higher economic and social status have a significant impact on children’s psychological resilience anxiety and depression, and mobile phone dependence. Being from *dibao* families has a direct effect on children’s fear of peers and inferiority; it also indirectly leads to a higher level of fear of peers and inferiority by affecting their psychological resilience. Thus, Hypothesis 4 was supported, but Hypothesis 6 and Hypothesis 8 were not. Compared to previous research [31,36,37,38], this study further found that urban–rural distinction has a significant impact only on children’s anxiety and depression, and mobile phone dependence.

Finally, this study found that, aside from urban or rural household registration and being from *dibao* families, children’s personal characteristics, family environment, and school environment all affect their fear of peers and inferiority in peer interactions. Specifically, as the children grow older, their fear of peers and inferiority become more serious. Girls face more serious fear of peers and inferiority than boys. Children who are not an only child have a higher level of fear of peers and inferiority than those who are an only child. Children’s health status is positively correlated with fear of peers and inferiority. Student leaders can help children diminish fear of peers and inferiority. Children from divorced families have more severe fear of peers and inferiority. Parental relationship, family relationship, and parent-child relationship are all negatively correlated with children’s fear of peers and inferiority. Children in public schools have a higher level of fear of peers and inferiority than those in private schools. Children in boarding schools have a higher level of fear of peers and inferiority than those in day schools. Children’s relationship with their teachers has a significantly negative correlation with fear of peers and inferiority.

### 5.2. Policy Suggestions

Positive peer interactions and developing and maintaining good peer relationships are conducive to children’s healthy growth. The government, family, and school should pay active attention to the problem of children’s fear of peers and inferiority, warranting timely interventions and help. They should encourage children to actively participate in peer interactions and create and maintain good peer relationships. Therefore, based on the research findings, this study provides the following suggestions.

First, policy makers should focus on the fear of peers and inferiority of rural children and children from *dibao* families. For rural children, it is necessary to concentrate on strengthening their psychological resilience, alleviating their anxiety and depression, and diminishing their dependence on mobile phones. For children from *dibao* families, it is essential to enhance their psychological resilience and prevent the *dibao* family from creating feelings of inferiority in their children’s social interactions.

Second, policy makers should pay special attention to fear of peers and inferiority in exceptional children (*teshuertong*). Psychological changes of social cognition in older children are worth our attention, and it is necessary for us to solve the problem of fear of peers and inferiority. More support for and attention to girls and children who are not an only child is needed to enhance their peer interactions. Policy makers should work to improve children’s health status to avoid the psychological problem of fear of peers and inferiority caused by health problems. Children from divorced families also deserve our attention, and we can help them by actively providing psychological support and encouraging them to engage in peer interactions to offset the negative psychological effects from divorce.

Third, families and schools should play important roles in solving the problem of fear of peers and inferiority. In a family, a harmonious conjugal relationship, supportive family atmosphere, and a close parent-child relationship are conducive to addressing children’s problem of fear of peers and inferiority. Parents are advised to have fewer quarrels or avoid them. We suggest that parents should often organize family outings and regularly communicate with their children. For schools, children living on campus are the center of attention, and each school should support them in engaging in peer interactions. It is also necessary to build and maintain good relationships between teachers and students, and teachers should guide children and become role models. Teachers are advised to encourage children to actively participate in peer interactions. Children should be motivated to play an active role in class activities and to campaign for student leaders. In doing so, they can improve their ability to avoid fear of peers and inferiority.

## Figures and Tables

**Figure 1 healthcare-10-02057-f001:**
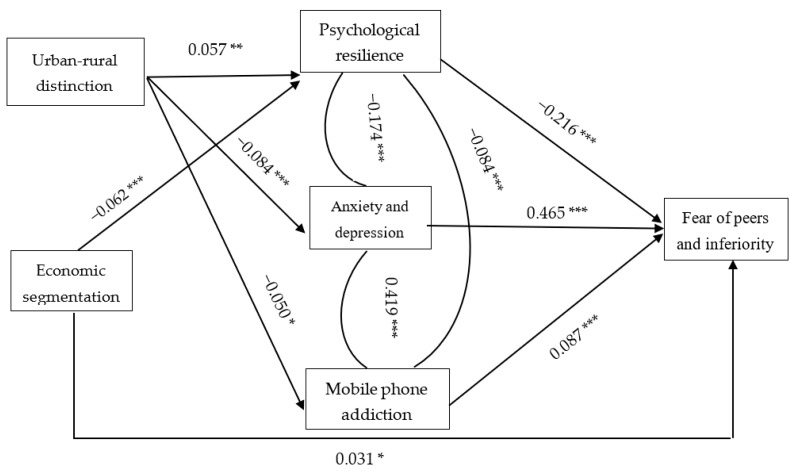
Model path diagram based on the standardized regression coefficients. *** *p* < 0.01, ** *p* < 0.05, * *p* < 0.1.

**Table 1 healthcare-10-02057-t001:** Differences in the characteristics of the two group of participants.

Variable Type	Variable	All Samples		Urban vs. Rural					*Dibao* Family vs. Non-*dibao* Family				
				Urban Samples		Rural Samples		*p*-Values	*Dibao* Family Samples		Non-*dibao* Family Samples		*p*-Values
		Mean	Std. Dev.	Mean	Std. Dev.	Mean	Std. Dev.		Mean	Std. Dev.	Mean	Std. Dev.	
Dependent variable	Fear of peers and inferiority	19.942	5.723	19.417	5.803	20.746	5.504	0.000	20.381	5.800	19.570	5.632	0.000
Independent variables	Economic segmentation	0.459	0.498	0.489	0.500	0.412	0.492	0.000	—				
	Urban–rural distinction	0.605	0.489	—					0.646	0.479	0.571	0.495	0.000
Mediating variables	Psychological resilience	104.473	17.539	106.120	17.421	101.950	17.426	0.000	102.503	17.953	106.141	17.009	0.000
	Anxiety and depression	42.882	9.371	41.926	9.270	44.347	9.340	0.000	43.141	9.340	42.663	9.395	0.000
	Mobile phone addiction	34.611	10.311	34.048	10.340	35.528	10.204	0.000	34.832	10.401	34.433	10.238	0.160
Covariates	Key school	0.105	0.306	0.125	0.331	0.073	0.260	0.000	0.091	0.288	0.116	0.320	0.010
	Public school	0.916	0.278	0.926	0.262	0.900	0.300	0.005	0.928	0.258	0.905	0.293	0.008
	Boarding school	0.211	0.408	0.124	0.329	0.344	0.475	0.000	0.211	0.408	0.211	0.408	0.499
	Whether parents are alive	0.871	0.335	0.869	0.337	0.874	0.332	0.355	0.795	0.404	0.934	0.248	0.000
	Whether parents are divorced	0.166	0.372	0.188	0.390	0.132	0.339	0.000	0.224	0.417	0.117	0.321	0.000
	Parents’ quarrel	0.281	0.450	0.278	0.448	0.287	0.453	0.278	0.270	0.444	0.291	0.454	0.089
	Family gathering	1.903	1.482	2.126	1.490	1.563	1.405	0.000	1.694	1.499	2.081	1.445	0.000
	Parent-child communication	0.785	0.411	0.804	0.397	0.758	0.429	0.000	0.742	0.437	0.822	0.383	0.000
	Parents’ beating and scolding	0.092	0.289	0.091	0.288	0.093	0.291	0.420	0.082	0.275	0.100	0.300	0.039
	Gender	0.499	0.500	0.513	0.500	0.477	0.500	0.021	0.517	0.500	0.483	0.500	0.028
	Only child	0.364	0.481	0.480	0.500	0.187	0.390	0.000	0.387	0.487	0.345	0.476	0.007
	Health status	0.953	0.211	0.953	0.211	0.953	0.212	0.480	0.926	0.261	0.976	0.154	0.000
	Physical disability	0.022	0.147	0.021	0.143	0.024	0.152	0.303	0.028	0.166	0.017	0.128	0.012
	Student leader	0.423	0.494	0.459	0.498	0.368	0.482	0.000	0.378	0.485	0.461	0.499	0.000

Notes: *t*-tests were used for continuous variables, and proportion tests were used for variables in proportions.

**Table 2 healthcare-10-02057-t002:** Analysis results of the multiple regression models.

Variable	Model 1	Model 2	Model 3	Model 4
Urban–rural distinction	−1.399 ***	−1.173 ***	−0.966 ***	−0.681 ***
	(0.199)	(0.210)	(0.212)	(0.215)
Economic segmentation	0.916 ***	0.897 ***	0.708 ***	0.573 ***
	(0.198)	(0.199)	(0.204)	(0.203)
Key school		−0.116	−0.102	−0.0102
		(0.333)	(0.329)	(0.331)
Public school		0.729 **	0.671 *	0.846 **
		(0.349)	(0.357)	(0.356)
Boarding school		1.074 ***	0.965 ***	0.934 ***
		(0.242)	(0.246)	(0.244)
Whether parents are alive			0.119	0.0383
			(0.312)	(0.310)
Whether parents are divorced			0.428	0.616 **
			(0.288)	(0.292)
Parents’ quarrels			1.256 ***	1.162 ***
			(0.220)	(0.218)
Family gatherings			−0.286 ***	−0.261 ***
			(0.0707)	(0.0699)
Parent-child communication			−1.036 ***	−0.955 ***
			(0.255)	(0.255)
Parents’ beating and scolding			1.630 ***	1.651 ***
			(0.368)	(0.364)
Gender				−0.867 ***
				(0.195)
Only child				−0.547 **
				(0.222)
Health status				−0.584
				(0.509)
Physical disability				0.944
				(0.687)
Student leader				−1.290 ***
				(0.199)
Constant	20.37 ***	19.36 ***	20.08 ***	21.48 ***
	(0.171)	(0.379)	(0.536)	(0.746)
Observed value	3334	3303	3240	3238
R2	0.019	0.025	0.058	0.078

Robustness standard errors are reported in parentheses. *** *p* < 0.01, ** *p* < 0.05, * *p* < 0.1.

**Table 3 healthcare-10-02057-t003:** Estimation results of the structural equation model.

	Psychological Resilience	Anxiety and Depression	Mobile Phone Addiction	Fear of Peers and Inferiority in Peer Interactions		
	Direct	Direct	Direct	Direct	Indirect	Total Effect
Urban–rural distinction	2.040 ***	−1.609 ***	−1.061 **	0.359	−0.652 ***	−0.689 **
	(0.630)	(0.403)	(0.432)	(0.164)	(0.126)	(0.214)
Economic segmentation	−2.180 ***	0.163	0.046	−0.037 **	0.202 *	0.561 ***
	(0.594)	(0.328)	(0.406)	(0.174)	(0.119)	(0.201)
Psychological resilience	-	-	-	−0.070 ***	-	−0.070 ***
				(0.005)		(0.005)
Anxiety and depression	-	-	-	0.284 ***	-	0.284 ***
				(0.010)		(0.010)
Mobile phone addiction	-	-	-	0.048 ***	-	0.048 ***
				(0.010)		(0.010)
Key school	3.143 ***	−0.115	−0.145	0.320	−0.261	0.059
	(0.929)	(0.513)	(0.625)	(0.256)	(0.186)	(0.315)
Public school	0.050	−0.260	0.228	0.922 ***	−0.066	0.855 **
	(1.041)	(0.575)	(0.704)	(0.286)	(0.207)	(0.353)
Boarding school	−0.525	0.754 *	1.999 ***	0.604 ***	0.348 **	0.952 ***
	(0.750)	(0.415)	(0.511)	(0.207)	(0.150)	(0.255)
Whether parents are alive	−0.644	−0.582	−0.807	0.147	−0.159	−0.012
	(0.879)	(0.485)	(0.603)	(0.242)	(0.175)	(0.298)
Whether parents are divorced	−0.860	1.857 ***	1.467 ***	−0.014	0.659 ***	0.645 **
	(0.788)	(0.435)	(0.548)	(0.218)	(0.158)	(0.267)
Parents’ quarrel	−4.333 ***	2.581 ***	1.579 ***	0.052	1.115 ***	1.166 **
	(0.638)	(0.351)	(0.433)	(0.178)	(0.130)	(0.216)
Family gathering	1.460 ***	−0.502 ***	−0.323 **	−0.010	−0.261 ***	−0.272 ***
	(0.199)	(0.110)	(0.138)	(0.055)	(0.040)	(0.068)
Parent-child communication	1.460 ***	−1.086 ***	−0.183	−0.040	−0.858 ***	−0.898 ***
	(0.199)	(0.394)	(0.503)	(0.200)	(0.147)	(0.242)
Parents’ beating and scolding	−5.602 ***	5.477 ***	3.631 ***	−0.470 *	2.126 ***	1.656 ***
	(0.990)	(0.545)	(0.684)	(0.277)	(0.203)	(0.335)
Gender	−0.951 *	−0.289	2.050 ***	−0.958 ***	0.084	−0.874 ***
	(0.568)	(0.314)	(0.387)	(0.158)	(0.115)	(0.193)
Only child	1.297 **	−1.196 ***	0.212	−0.161	−0.421 ***	−0.582 ***
	(0.629)	(0.347)	(0.427)	(0.174)	(0.126)	(0.213)
Health status	2.063	−1.496 **	0.886	0.036	−0.527 *	−0.491
	(1.358)	(0.750)	(0.985)	(0.375)	(0.272)	(0.461)
Physical disability	−2.957	−0.053	−2.299	0.830	0.082	0.912
	(1.948)	(1.075)	(1.404)	(0.537)	(0.389)	(0.661)
Student leader	5.174 ***	−0.222	−0.928 **	−0.838 ***	−0.472 ***	−1.311 ***
	(0.583)	(0.321)	(0.395)	(0.162)	(0.119)	(0.198)

Standard errors are reported in parentheses. *** *p* < 0.01, ** *p* < 0.05, * *p* < 0.1.

## Data Availability

The data that support the findings of this study are available from the corresponding author upon reasonable request.

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
