# Peer review of "Urban–Rural Distinction or Economic Segmentation: A Study on Fear and Inferiority in Poor Children’s Peer Relationships"

_healthcare, 2022, doi:10.3390/healthcare10102057_

Round 1

Author Response

Response to  Comments

This paper used a unique and nationally representative dataset of poor children in China and investigated the effects of urban-rural distinction and economic segmentation on poor children’s fear and inferiority in peer interaction. It also explored the specific mechanisms on how these effects took place by examining the mediating roles psychological resilience, anxiety and depression, and mobile phone dependence played in affecting the above relationships. In my opinion, this paper is a quite interesting and well-organized study, making important contribution to child development theories and policy making. I think the paper would be improved by addressing the following issues.

Response: Thank you for your thorough reading of our paper, and for acknowledging our contribution to the field. We highly appreciate your recognition of our efforts and the care you have taken with our paper and the suggestions offered. We address each of your comments below.

  1. On “1. Background”. The authors may present the general background, previous studies, and the focus of this paper in different paragraphs instead of in one paragraph.

Response: Thank you for this comment. We renamed “1. Background” as “1. Introduction” and presented the general background, previous studies, and the focus of this paper in different paragraphs in this revised manuscript. Please see the “1. Introduction” for more information.

  1. On “2. Research questions and analysis frameworks”. The authors may clarify their research questions at the beginning of this section. When stating the mediating effects of the three mediators, they may first present the effects of predictors (urban- rural distinction and economic segmentation) on the mediators. When proposing the hypotheses of the mediating effects, they may reorganize the language. For example, “Children’s psychological resilience mediates the effect of urban-rural distinction on their fear and inferiority in peer interaction. Specifically, living in rural areas decreases children’s resilience level, which in turn increases their fear and inferiority in peer interaction”.

Response: Thank you for the invaluable comments. We revised the section based on your suggestion. Please see “2. Research Questions and Analytical Frameworks” for more information.

  1. On “3. The approach”. The authors may change the title of the section as “Methods”. They may also clarify the definitions of some control variables, such as key school, boarding school, family outings, and so on. They may clarify the analytical strategies by moving the structural equation modeling design from “Results” (lines 366-374) to “Tools and model” (lines 226-232)

Response: Thank you for the invaluable comments. We revised the section based on your suggestion. Please see “3. Methods” for more information.

  1. On “4. Descriptive analysis and analysis results of regression model”. The authors may change the title of the section as “Results”. They may exclude some repeated statements on descriptive results (lines 235-261) and regression results (lines 276- 362).

Response: Thank you for the invaluable comments. We revised the section based on your suggestion. Please see “4. Results” for more information.

  1. The authors may replace some references of papers from Chinese journals with those from international peer-reviewed journals.

Response: Thank you for the invaluable comments. We revised the section based on your suggestion. Please see the new references for more information.

  1. The authors should ask for a professional editor to improve their writing, especially the sentence structure.

Response: Thank you for the invaluable comments. We asked a professional editor for language editing. Please see the revised manuscript.

Reviewer 2 Report

The background should have some empirical evidence. Research should have a good aim and Objectives indicated. Hypothesis should be generated followed by research questions. Hypothesis genratio should be readdress with the style of word combination and it should be very clear to understand. If you consider the multiple regression anlysis it should be presented in a sensible manner. 

Author Response

Responses to comments

The background should have some empirical evidence. Research should have a good aim and Objectives indicated. Hypothesis should be generated followed by research questions. Hypothesis genratio should be readdress with the style of word combination and it should be very clear to understand. If you consider the multiple regression anlysis it should be presented in a sensible manner. 

Response: Thank you for your thorough reading of our paper. We highly appreciate your recognition of our efforts and the care you have taken with our paper and the suggestions offered. We address each of your comments below.

  1. Background can have more empirical evidences in terms of literatures.

Response: Thank you for this comment. We renamed “1. Background” as “1. Introduction” and revised the section based on your comment. Please see the “1. Introduction” for more information.

  1. Research questions could have been generated first and subsequently the hypothesis.

Response: Thank you for the invaluable comments. We revised the section based on your suggestion. Please see the revision for more information.

  1. Please readdress the way of writing up the hypothesis.

Response: Thank you for the invaluable comments. We revised the section based on your suggestion. Please see “2. Research Questions and Analytical Frameworks” for more information.

  1. This table is not essentially appropriate. Instead, better to show sample items in the instruments sections.

Response: Thanks for the comment. We revised the manuscript based on your comment. Please see the new Table 1.

  1. Please indicate summary of multiple regression results.

Response: Thanks for the comment. We rewrote the results section. Please see “4. Results” for more information.

Reviewer 3 Report

See attachment

Author Response

Responses  to comments

Using national survey data in China, authors analyze two behavioral traits in children: fear and inferiority complex, focusing on the potential influence of one’s social status (rural households and low-income households). Other dimensions evaluated include cell phone use, anxiety and depression, and resilience. Authors ‘find’ evidence that isolation effects are more likely among children from low-income and rural households relative to their urban and wealthy counterparts, and that this social class divide also has indirect negative effects on fear and inferiority. The topic is a good one but authors attempt to do too much in this paper but may have actually ended up doing far worse than they believe. The methods are inconsistent with their data and this makes the analysis inconsistent with their research questions!

Response: Thank you for your thorough reading of our paper. We highly appreciate your recognition of our efforts and the care you have taken with our paper and the suggestions offered. We address each of your comments below.

In the abstract, provide a complete and brief description of how your regression and structural equation models are used in the paper.

Response: We revised the Abstract in this revised manuscript. Please see the new Abstract for more information.

In section 3.2, please identify the range values for the total “score of peer fear and inferiority.” On Table 2, you have maximum and minimum values of 40 and 10 respectively for this variable, is this the whole range of values possible for this total “score of peer fear and inferiority”? How is it calculated? As weighted average? Same argument for other variables such as “psychological resilience”, “anxiety and depression” and “mobile phone dependence.”

Response: Thanks for the comment. We revised the section based on your comment. Please see “3.2. Measurements” for more information.

What is your measure of “low-income”? Is it represented by “whether the child is from a family receiving basic living allowances”? If so, you may want to reframe your analysis as “poor households” rather than “low-income households.” All “poor households” will receive basic living allowances but not all “low-income households” will receive these basic living allowances. Without this change, you must distinguish what it means to be poor vs to be low-income.

Response: Thanks for the comment. We are sorry for the ambiguous and inconsistent terms. We changed “low-income family” to “poor (dibao) family”. Please see the revised manuscript for more information.

On Table 1, at least 7 of the variables listed are categorical variables, labelled with values ranging from 1 to 5. Are these values what authors use in their regressions? If so, the whole analysis is a worthless exercise. It is not how regressions work.

Response: Thanks for the comment. We have revised the multinomial categorical variables to multiple dichotomous variables and re-ran all the models in this revised manuscript.

The discussion in section 4 includes references to “t-test.” Where do you report these tests in the paper? And what is even its importance in the paper. Same question about Table 3!

Response: Thanks for the comment. We added the T-test results in this revised manuscript. Please see the new Table 1 for more information.

Identify the dependent variable when you report regressions results.

Response: Thank you for the comment. We rewrote the results section. Please see the revised manuscript.

Some of the variables in the model should be treated as interactions rather than independently. Also, the rural-urban distinction together with low-income and high- income classification seems inappropriate. Are there no low-income urban households in China?

Response: Thanks for the comment. We agree that including interaction terms may make the study more interesting, but we finally decided not to include them for two reasons. First, this study focuses more on the independent effects of the two independent variables; and second, given the current is already very complex, it may not be appropriate to add more complexities. Thanks for your understanding.

Round 2

Reviewer 2 Report

Dear Authors,

Please revisit a bit the wording of peer fear as fear of peers and hypothesis intepretations in the discussion part. 

Author Response

Please revisit a bit the wording of peer fear as fear of peers and hypothesis interpretations in the discussion part.

Response: Thanks for this comment. We replaced “peer fear” with “fear of peers” in this revised manuscript. We also revised the hypothesis interpretations in the discussion. Please see the manuscript for more information.

Here the Dibao families are poor families. The comparison also mentioned about Dibao families as well.

Response: Thanks for this comment. We are sorry for the error. We mean the comparison between Dibao and non-Dibao families. We have corrected the mistake in this manuscript.

Reviewer 3 Report

Authors did their best to address the issues raised in the first review. In addition to further editing the paper for typos and grammar, there is a curious element on Table 1. Specifically, the data on some of the variables appears to be in proportions. Thus, t-tests are inappropriate. Use proportions test.

Author Response

Authors did their best to address the issues raised in the first review. In addition to further editing the paper for typos and grammar, there is a curious element on Table 1. Specifically, the data on some of the variables appears to be in proportions. Thus, t-tests are inappropriate. Use proportions test.

Response: Thanks for this comment. T-tests were used for continuous variables, and proportion tests were used for variables in proportions. We re-ran all the analyses and report the new results in this manuscript. Please see Table 1 for more information.